# Biotechnology Advances in Bioremediation of Arsenic: A Review

**DOI:** 10.3390/molecules28031474

**Published:** 2023-02-03

**Authors:** Jaganathan Sakthi Yazhini Preetha, Muthukrishnan Arun, Nandakumar Vidya, Kumaresan Kowsalya, Jayachandran Halka, Gabrijel Ondrasek

**Affiliations:** 1Department of Biotechnology, Bharathiar University, Coimbatore 641006, Tamil Nādu, India; 2Faculty of Agriculture, The University of Zagreb, Svetosimunska c. 25, 10000 Zagreb, Croatia

**Keywords:** arsenic, microbial bioremediation, phytoremediation, genetic modification

## Abstract

Arsenic is a highly toxic metalloid widespread in the Earth's crust, and its contamination due to different anthropogenic activities (application of agrochemicals, mining, waste management) represents an emerging environmental issue. Therefore, different sustainable and effective remediation methods and approaches are needed to prevent and protect humans and other organisms from detrimental arsenic exposure. Among numerous arsenic remediation methods, those supported by using microbes as *sorbents* (microbial remediation), and/or plants as *green factories* (phytoremediation) are considered as cost-effective and environmentally-friendly bioremediation. In addition, recent advances in genetic modifications and biotechnology have been used to develop (i) more efficient transgenic microbes and plants that can (hyper)accumulate or detoxify arsenic, and (ii) novel organo-mineral materials for more efficient arsenic remediation. In this review, the most recent insights from arsenic bio-/phytoremediation are presented, and the most relevant physiological and molecular mechanisms involved in arsenic biological routes, which can be useful starting points in the creation of more arsenic-tolerant microbes and plants, as well as their symbiotic associations are discussed.

## 1. Introduction

Contamination of soils and waters with arsenic (As) as a highly toxic element is an increasing environmental issue. The safety level of As in water (10 ppb) and soil (100 mg/kg) [1] has already been exceeded by multifold (e.g., 50 ppb in drinking water) in many Asian countries, leading to various negative health implications, including carcinogenic [2,3]. Namely, As is classified as a Class 1 carcinogen element by the International Agency for Research on Cancer [4,5]. According to WHO, 300 million people are affected by As contamination of groundwater [6], while, only in Bangladesh, nearly 43,000 deaths are attributed to chronic As exposure every year [7]. The main causes of As contamination are different natural or anthropogenic processes and sources (Figure 1). Natural sources of As contamination include hydrothermal activity, volcanic emissions, ore deposits and Cenozoic sediments, whereas anthropogenic activities include exploitation of geothermal energy, industrial emissions, mining and excessive pumping of near-surface groundwater [8,9] (Figure 1). As contamination of water is mainly caused by the dissolution of rocks, minerals or ores, and by industrial effluents, including mining waste. In the soluble form, As(V) is usually present in oxygenated water such as wells, while As(III) is found in reducing conditions such as groundwater or deep lake sediments [1,10,11]. Both inorganic forms of As have been found in soils as well, and can be transported in the food chain by plant uptake and food crop consumption, inducing different toxic effects [12]. As(III) has been reported to be more toxic to humans than organic As and As(V) [13]. The main sources of As exposure for humans are the use of contaminated water for drinking/cooking, growing of crops in contaminated soils and/or irrigation with contaminated water and consumption of contaminated sea food [2,7,14]. Once As has entered the human body, it can cause a wide range of acute and chronic toxic effects and diseases, with or without specific symptoms, depending on As exposure, dosage and chemical form. For instance, symptoms of As exposure and poisoning include fatigue, colitis, loss of reflexes, weight loss, weakness, anorexia, gastritis, hair loss, anorexia, etc., while As-associated diseases and disorders are related to liver, kidney, cardiovascular, nervous and peripheral vascular function, hyperkeratosis and hyperpigmentation [6,12,13] (more in Section 3). In many experimental and clinical studies, it was confirmed that As exposure is a significant risk factor for many intellectual and neurological disorders, especially in the most vulnerable child population, where, even at a dose <10 μg/L, As can negatively impact memory and intelligence performances [6]. Finally, as a carcinogenic element, As exposure can trigger numerous types of cancer, including lung, breast, skin, liver, kidney and bladder cancer [3,4,6] (Section 3). Therefore, effective, environmentally-friendly and cost-efficient remediation strategies must be developed and applied to limit As contamination and protect humans from harmful As exposure.

Some of the most effective methods for remediation of As-contaminated soils and waters, such as (i) microbial bioremediation, i.e., using of microorganisms to detoxify metals from the environment, and/or (ii) phytoremediation, i.e., using of plants to detoxify or remove metals from the environment, have been discussed. In addition, the most recent insights related to molecular mechanisms in As microbial and phytoremediation, as well as plant–microbe associations and interactions at the rhizosphere level important in As remediation have been critically presented and reviewed. Finally, continued advances in biotechnology and materials science (e.g., nano-/bio-based materials, genetically modified organisms) are expected to enable the development of novel tools, techniques and products for monitoring and modelling crucial biogeochemical interactions and processes with As, which will ensure more efficient As remediation.

## 2. Arsenic Chemistry

Three different forms of As compounds occur in nature: organic, inorganic and arsenic gas [4]. Organic As is covalently bonded to As atoms and has a carbon atom in its structure, while the inorganic As form has no carbon atom in its structure and is a pure metalloid bonded to elements. Organic forms of As are less toxic in nature than inorganic As forms. In general, inorganic As exists in four different oxidation states: As^3-^ (arsine), As^0^ (elemental arsenic), As^3+^ (arsenite) and As^5+^ (arsenate) (Figure 2) [1,10,15,16]. Of these, arsenate (HAsO_4_^2-^, H_2_AsO_4_^−^) and arsenite (As(OH)_3_, HAsO_3_) are most abundant in soils and waters under oxic and anoxic conditions [1,10,15].

Organic forms of As are methylarsine [CH_3_AsH_2_], dimethylarsine [(CH_3_)_2_AsH], trimethylarsine [(CH_3_)_3_As], monomethylarsonic acid [CH_3_As(OH)_2_, MMA(III)], monomethylarsonic acid [CH_3_AsO-OOH], MMA(V)], dimethylarsinic acid [(CH_3_)_2_AsOH, DMA(III)], dimethylarsinic acid [(CH_3_)_2_AsOO-], DMA(V)], trimethylarsinic oxide [(CH_3_)_3_AsO, TMAO] and tetramethylarsonium ion [(CH_3_)_4_As^+^, TMA^+^] [17,18] (Figure 3).

## 3. Environment Arsenic Exposure and Toxicological Effects in Humans

Human exposure to As causes various diseases (skin lesions and keratosis) [19] and organ (lung, kidney, liver) failure [20]. The physiological effects of chronic As exposure include cardiovascular, renal, neurological, respiratory, reproductive, haematopoietic and haematological dysfunctions [21]. In addition, As causes genotoxic effects, including chromosomal aberrations, micronuclei formation, aneuploidy, deletion mutations, sister chromatid exchange and DNA-protein cross-linking [22]. Arsenicosis, i.e., waterborne disease caused by increased levels of As in drinking water, depends on the level and duration of exposure, genetic predisposition nutritional status and other factors [23]. The inorganic arsenic that enters human metabolism undergoes biochemical reactions such as oxidation, reduction, thiolation, glutathiolation and methylation [24]. In the oxidation-reduction reactions, As in humans reacts with oxygen radicals and can be catalyzed by endogenous reducing agents such as ascorbate, glutathione and tocopherol, depending on pH and the presence of other organometallic compounds to form As complexes [25]. Arsenic detoxification is catalyzed by SAM (S-adenosylmethionine methyltransferase), which converts inorganic As into methylarsonic acid (MMA) and dimethylarsinic acid (DMA) [26]. In the study of [27], it was concluded that epigenetic changes through over-methylation of CpG sequences and changes in DNA methylation serve as the basis for As carcinogenesis. It is reported that about 200 enzymes involved in cellular energy fluxes, DNA synthesis and repair have been inhibited under As exposure in humans [28]. In animal metabolism, trivalent arsenic (As(III)) binds to the sulphydryl group of the dihydrolipoamide, inhibiting pyruvate dehydrogenase and impairing ATP production, gluconeogenesis and other cellular cycles. In contrast, inorganic pentavalent arsenic (As(V)) replaces the phosphate group in the glycolytic and cellular respiratory pathways and forms ADP-arsenate instead of ATP [29].

## 4. Molecular Mechanisms Involved in As Microbial Remediation

The detoxification of As by microbes has been well documented in recent decades and plays an important role in bioremediation strategies [30,31]. Microbes can stabilize and detoxify heavy metals through enzymatic reduction reactions to oxidoreductases, making them less harmful to the environment [32]. The redox reactions of various As-resistant microbes isolated from different environmental samples act mainly on As(III) and As(V), converting inorganic to organic states [33]. The microbes also colonize all ecological niches and survive in extreme environments with immense gene reserves. This provides insight into the application of biotechnological approaches that use genomics for arsenic remediation with potential benefits [34]. The detoxification or accumulation of As involves different metabolic processes such as oxidation, reduction, biosorption, methylation and volatilization [35] (Figure 4).

### 4.1. Oxidation and Reduction of As

The oxidation of As is mainly mediated by chemicals and microbes in the environment [36]. The release of As into the environment or groundwater is triggered by the oxidation of mineral ores by reductive dissolution, e.g., As-containing sulphides, As-containing rocks and iron oxides [37]. The conversion of As(III) to As(V) by oxidation is generally considered to be effective bioremediation, as As(III) is significantly more (i.e., by an order of magnitude) toxic than As(V). Arsenite oxidase (*AioBA*), an enzyme responsible for catalysis of arsenite oxidation, was first purified and characterized from *A. faecalis* [38]. Arsenite oxidase, synthesized by the bacterial species *Thermus thermophiles*, *Thermus aquaticus*, *Crysiogenes arsenates*, *Bacillus arsenic oselenatis*, *P. arsenitoxidans*, *Desulfutomaculum auripigmentu*, *Geospirillum barnesi* and *Geospirillum arsenophilus*, oxidizes As(III) to As(V). Similarly, As(V) can also be reduced to As(III) by dissimilatory reduction during anaerobic respiration [39]. The regulation of As(III) oxidation is reviewed in the bacterial system [40]. There, the Aiox senses a signal from As(III) when it is present in the bacterial periplasm, which induces autophosphorylation to initiate the *AioBA* gene response (Figure 4).

The microbes take up As(III) and transport it via glycerol transporters such as aquaglyceroporins and As(V) via phosphate transporters [41] (Figure 4). As^5+^, as a potential inhibitor of oxidative phosphorylation, interrupts the metabolic reactions of phosphorylation and inhibits ATP synthesis by entering the Pit and Pst systems of *E. coli*, as it has a structural similarity to phosphate [15]. In *E. coli*, the uptake of As^5+^ is facilitated by these two transporters, with Pit being the main mediator of the layer [14]. The uptake of arsenate in organisms occurs under two conditions: (i) arsenate uptake in the presence of phosphate deficiency and (ii) adventitious uptake of arsenate by phosphate transporters. In *E. coli*, Pst has a high affinity and low capacity for arsenate, while Pit is a constitutive system with low affinity and high capacity for arsenate uptake [42]. The oxidation and reduction process of arsenate in *E. coli* and *S. cerevisiae* is mediated by Pit and Pho87p, phosphate transport membranes, in response to arsenate uptake and resistance in the cytoplasm. The cytoplasmic arsenate reductases ArsC and Arr2p reduce As(V) to As(III), and the reduced As(III) is transported by the glycerol transporters GlpF and Fps1p to remove arsenite from the cytoplasm [43]. The reduced As(III), which is sometimes sequestered in the cellular compartments, serves either as a conjugate for GSH or thiols or as free arsenite [44]. Some of the As-tolerant microbial strains are listed in Table 1, which provides insight on how to effectively bioremediate As or manipulate these genetic traits for potential benefits.

### 4.2. Methylation of As

Microbes such as bacteria, fungi and archaea play an important role in the conversion of inorganic arsenate to an organic state in the geochemical methylation cycle [56] (Figure 4). Methylation of arsenic by microbes is considered a detoxification pathway in aerobes that converts methylated arsenite (more toxic) to methylated arsenate (less toxic) [57]. This process is catalyzed by the enzyme As(III) S-adenosylmethionine methyltransferase in microbes, which is encoded by the *arsM* genes [35]. The mechanism of reduction of arsenate to trimethylarsine in *S. brevicaulis* by SAM (S-adenosylmethionine methyltransferase), an enzymatic methyl donor, involves seven sequential steps of conversion of arsenate to the final product trimethylarsine with the intermediates arsenite, methylarsonate, dimethylarsinate, dimethylarsinite and trimethylarsine oxide [58]. The conversion of this methylated end product trimethylarsine is less toxic and volatilizes in the environment [59].

### 4.3. Operon Expression in As Resistance

In bacteria, the arsenic resistance system (*ars*) operon is located in chromosomes or plasmids. The *ars* genes function as detoxifiers of inorganic arsenic compounds. In the detoxification process of As(III), the extrusion of As(III) is catalyzed by the arsenite efflux permease (*Ars B*), which is encoded by the *ars B* gene of the *ars* operon [60]. Expression analysis of As-resistant multi-operons (*ars*) in the arsenic detoxification process of *R. palustris* showed that the expression of *ars2* and *ars3* increased with increasing As(lll) concentration in the environment [61]. The *ars* operon cloned from the As-resistant microbe *A. multivorum* to overexpress *arsD*, *arsA* and *arsC* was expressed in *E. coli* and conferred resistance to arsenate and arsenite [62] (Figure 4).

### 4.4. Biosorption and Bioaccumulation of As

Biosorption and bioaccumulation by microbes is considered an effective method of bioremediation of toxic metals and metalloids. Biosorption of As in microbes is an energy-dependent process involving chelation, ion exchange and physical absorption via physicochemical interactions (Figure 4). The process of biosorption by microbes such as bacteria, algae, fungi and yeasts is mediated by the binding of As to different functional groups such as -COOH, -SH, -OH, -NH_2_ and -PO_4_^−3^ [63]. Bioaccumulation is a process of sequestering free As transported by glycerol and phosphate transporters by binding to proteins or peptides to reduce its toxicity [63].

## 5. Molecular Mechanisms Involved in As Uptake by Plants—Phytoremediation

Plants take up heavy metals dominantly via the root system [64] and a series of molecular mechanisms [65], and eventually detoxify the heavy metals through defense strategies and mechanisms [66] (Figure 5). The potential benefits of plants as hyperaccumulators of As can be further enhanced by direct overexpression of the gene involved in the phytoremediation process of metabolism, uptake or transport of heavy metals [67]. The uptake of As in plants depends on the plant species and its affinity to As. The inorganic forms of arsenite and arsenate that enter plant cells disrupt their normal metabolism by generating ROS such as superoxide radicals (O^2−^), hydroxyl radicals (OH) and hydrogen peroxide (H_2_O_2_) in response to the As detoxification process [39]. The methylated forms of As found in soil, monomethylarsinic acid (MMA) and dimethylarsinic acid (DMA), are also absorbed by plants [68]. Plant roots are capable of a rapid uptake of certain metall(oid)s [64], including As, which is then distributed throughout the plant tissue via the xylem [65]. The xylem accumulates pentavalent arsenic more than trivalent arsenic in plants [69]. Arsenite is transported through the aquaporin channel via nodulin26-like intrinsic proteins in the root cells, which are encoded by plant genes [70]. Arsenite disrupts the plant cell membrane by binding to protein sulphhydryl groups and causes cell death. During arsenite detoxification, the binding of arsenite to peptide-rich sulphydryl groups forms a complex with phytochelatins (As-PC) (Figure 5). In this form, As can be sequestered by ABC transporters in the cell vacuoles and transported via the phloem to the roots and stems, making it less toxic [71]. Arsenate, the phosphate analogue, is transported by the phosphate transporters (Pht 1:1 and Pht 1:4) in plant cells (Figure 5). This entry of arsenate interrupts phosphate by replacing it in various metabolic pathways and interrupts ATP synthesis [71]. In the detoxification process, arsenate (As(V)) is reduced to arsenite (As(III)) by the As reductase *ARS2* in the intracellular compartment. The reduced As(III) is detoxified by forming a complex with thiol-rich peptides or being effluxed out of the of the cell via glycerol transporters [5] (Figure 5).

For instance, in rice, arsenate enters through the phosphate transporters OsPHT1;1 and OsPHT1;8 and AtPHT1;1, AtPHT1;4, AtPHT1;5, AtPHT1;7, AtPHT1;8 and AtPHT1;9 in *A. thaliana*. The uptake and accumulation of arsenate is also mediated by Phosphate transporter (PHT) proteins and regulators, PHR2 (Phosphate starvation response 2) and OsPHF1 (Phosphate transporter traffic facilitator 1). On the contrary, arsenite enters rice via OsNIP1;1, OsNIP2;2, OsNIP3;1, OsNIP3;2 and OsNIP3;3. In *A. thaliana,* arsenite enters the aquaglyceroporins of AtNIP1;1, AtNIP1;2, AtNIP3;1, AtNIP5;1, AtNIP6;1 and AtNIP7;1 [72]. As(III) can be detoxified in cells by combining with phytochelatins to form a complex of As(III)-phytochelatins, which are then sequestered in vacuoles. The complex is then transported through the tonoplast by the C-type ATP-binding cassette (ABC) for detoxification [73] (Figure 5). The inorganic form of methylated arsenic, MMA and DMA, is transported in rice by the silica transporter (Lsi 1). DMA and MMA are taken up into the root cells by the protein NIP. DMA is more mobile than MMA and is readily translocated to other tissues via the xylem and phloem by interaction with the gene *OsPTR7* [74]. Some plant species with potential hyperaccumulation of As are listed in Table 2.

## 6. Biotechnological Strategies for Improvement of Microbial Bioremediation of As

Biotechnological strategies exploit the understanding of molecular determinants in the process of As remediation. Bioremediation of As with microbes by manipulating the genes for effective As accumulation and resistance has evolved over the past decades. For instance, an *arsM* gene, which encodes Cyt19 As(III) S-adenosylmethionine methyltransferase in *Rhodopseudomonas palustris*, is regulated by As compounds. This *arsM* gene was highly expressed in *Sphingomonas desiccabilis* and *Bacillus idriensis* [82]. In addition, it was confirmed that transgenic microbes with expressed *arsM* have the ability to remove As (e.g., 2.2 to 4.5% in the soil and up to 10-fold in nutrient solution) by biovolatilization compared to the wild type strains [82]. Similarly, in the study of [83], the thermophilic strain *Bacillus subtilis* 168, which is unable to methylate and volatilize As, was genetically modified by expressing the *CmarsM* gene (arsenite S-adenosylmethionine methyltransferase) from the heat-resistant alga *Cyanidioschyzon merolae*. The genetically modified strain, *Bacillus subtilis* 168 with expressed *CmarsM* gene, methylated As to dimethylarsenate and trimethylarsine oxide and volatilized dimethylarsine and trimethylarsine. This conversion of inorganic As to the methylated organic form and volatilization occurred within 48 h in As-contaminated organic compost when inoculated with the genetically modified strain. This study suggests the use of the genetically modified strain for bioremediation of As-contaminated compost. In the case of [84], PCR analyses of bacterial samples isolated from different As-contaminated zones confirmed that the diversity of bacterial arsenite transporters among the contaminated soil bacteria is low, whereas high diversity was found compared to isolates from low contaminated soils. It was also confirmed that 70.7% of the isolates from As-contaminated soils showed expression of the *Ars* gene family. This study suggests that the novel traits studied in these organisms could serve as potential approaches for As bioremediation. In the study conducted by [85], the As-resistant bacterium *Corynebacterium glutamicum* was genetically modified to improve the bioremediation of As. In *C. glutamicum*, As reduction is mediated by ArsC1 and ArsC2, mycothiol-based single-cysteine reductases. Here, the three-cysteine homodimer ArsC1, encoded by the *ars 1* operon, is constitutively expressed to increase As(V) uptake in this strain. In addition, the mutant *C. glutamicum* strains were able to accumulate 15-fold more As(lll) and 30-fold more As(V) in comparison to the control treatment. This study also suggests that *C. glutamicum* could serve as a bio-tool for effective bioremediation of As.

Previously, in study performed by [86], As accumulation in bacterial cells was enhanced by the expression of *ArsR*. In this experiment, overexpression of *ArsR* with elastin-like polypeptide (ELP153AR) increased arsenic accumulation in cells 60-fold compared to control cells. It also removed 100% of the arsenite (50 ppb) in the contaminated site, suggesting that genetically engineered *E. coli* cells with high *ArsR* affinity represent a cheaper and more effective method for future use in bioremediation strategies. In the study by [87], the regulation of two As-responsive repressors, *ArsR1* and *ArsR2* in *Pseudomonas putida* strain KT2440, was investigated with the corresponding promoters. The cross talk between the *ArsR1*/*ArsR2* repressors was investigated for their role as transcriptional inducers in arsenate and arsenite tolerance. In vitro examination of these genes revealed that each of the two promoters acts as a repressor of the other, corresponding to a key functional bifan motif. This study suggests that the coexistence of *ArsR* variants regulates evolutionary function as repressors of As tolerance.

## 7. Biotechnological Strategies for Improvement of Phytoremediation of As

Transgenic plants have been developed that can accumulate or detoxify arsenic from contaminated soils or waters for effective phytoremediation. In As-tolerant plants, genetic manipulation of arsenic resistance or detoxifying properties have been studied and analyzed in different conditions over the past decades and are discussed in the next paragraphs.

For instance, in the study performed by [88], a transgenic approach to arsenic hyperaccumulation was taken by genetic engineering, where *PvACR3*, an arsenite antiporter in the As-hyperaccumulating plant *Pteris vittate*, was expressed in *A. thaliana*. The transgenic plants in which *PvACR3* was expressed through the CAMV promoter showed tolerance to 80 µm arsenite and 1200 µm arsenate compared to the control, which is reported to be a lethal dose for wild-type *A. thaliana*. These genetically modified plants showed a 7.5-fold increase in As uptake compared to the wild types. In the experiment conducted by [89], *AtACR2*, the As reductase gene of *A. thaliana*, was cloned and expressed in the modified genome of tobacco to generate As-tolerant transgenic lines. The results of this study showed that transgenic tobacco plants with the *AtACR2* gene expressed increased As uptake in root tissue compared to wild-type tobacco plants. The plants also survived in a medium with a higher arsenic concentration of 200 µm arsenate, while the wild-type tobacco plants barely survived at this concentration. This study provides insight into the potential use of the *AtACR2* gene as a genetic tool for effective arsenic phytoremediation. In contrast, the accumulation of As in rice grains is still one of the major environmental concerns, as rice is a staple food for most people [90,91]. In the study of [92], As transporter OsPht1:8 and Lsi1/2 mutations in rice reduced the accumulation of As in rice plants and reduced its accumulation in rice grains. In the same way, in [93], mutations of the gene OsPT8 in two varieties of Nipponbare and Kasalath rice showed a 33–57% decrease in arsenate uptake compared to wild types and increased arsenate tolerance in the mutants. This shows that genetic traits can also be reversibly eliminated by a transgenic approach in phytoremediation for the benefit of the environment.

In the phytoremediation strategy of [94], the As reductase gene (*arsC*) of *E. coli* was expressed in transgenic *A. thaliana* along with the soybean Rubisco promoter (*SRS1p*) and the constitutive actin promoter (*ACT2p*), resulting in increased As accumulation in transgenic vs. wild-type *A. thaliana*. These transgenic plants with expressed *SRS1p*/*ArsC* and *ACT2p*/*ƴ-ECS* showed a 2- to 3-fold increase in As accumulation in shoots compared to the control. It was suggested that microbial gene expression with specific promoters could enhance the phytoremediation ability of each plant species. Furthermore, in a study by [95], transgenic *A. thaliana* was engineered by expressing the arsenic methyltransferase gene (*WaarsM*) of the fungus *Westerdykella aurantiaca* isolated from As-contaminated soil. Transgenic *A. thaliana* plants expressing *WaarsM* showed higher tolerance to arsenite and arsenite than control wild-type plants. When these transgenic plants were exposed to 50 µM arsenite and 250 µM arsenate for 48 h, they developed 113 ng and 17.5 volatile As compounds, respectively, while long-term exposure to As resulted in less As accumulation in transgenic plants than in wild-type plants. This study thus suggests that volatilization of transgenic plants expressing the *WaarsM* gene could be an effective new method for effective phytoremediation of As (e.g., Figure 4). In [96], the *CrarsM* gene of the eukaryotic alga *Chlamydomonas reinhardtii* was expressed in *A. thaliana* to methylate As for bioremediation. This study resulted in the production of *CrarsM*-expressing *A. thaliana* transgenic plants with high efficiency in the methylation of As(III) to DMA(V) and low volatilization. Transgenic plants expressing the *CrarsM* gene were also arsenic resistant compared to wild-type plants (As-sensitive). In the gene silencing study of *ACR2* [97], the only gene identified in *A. thaliana ACR2* that reduced the activity of arsenate to arsenite was blocked by RNA interference. This resulted in 10- to 16-fold higher arsenate uptake in shoots than in the wild type, suggesting that RNA interference of *ACR2* could enhance the phytoremediation of arsenate in many plant species.

## 8. Interaction of the Rhizosphere Microbial Community on the Phytoremediation of As

Microbes and plants play an important role in the detoxification or accumulation of heavy metals in soil through synergistic actions. Generally, in phytoremediation of heavy metals, plants follow strategies such as rhizodegradation, phytostabilization, rhizofiltration, phytoextraction, phytodegradation, phytoaccumulation and phytovolatilization [98]. Soil bacteria in metal-contaminated zones interact with the phytoremediating plants to improve plant metal tolerance, enhancing plant growth and altering accumulation of heavy metals in the plant tissues [99]. The microbes that take up the compounds from the root exudates can detoxify or reduce metals in the rhizosphere [100]. However, the interactions of soil microbiomes and plants in metal(oid)s remediation are poorly understood and explained, notably in the case of As. For example, a reduced natural microbiota was confirmed in the As-contaminated rhizosphere, with a gradual enrichment of microbial genes involved in As(III) oxidation, As(V) reduction and As (de)methylation in the rhizosphere, following increased As uptake by the hyperaccumulator *Pteris vittata* [101]. In the study by [102], the *arsM* gene was found to be abundant in the roots and rhizosphere of rice, suggesting that methylated As accumulated in rice may be the result of horizontal gene transfer from the soil microbiota. Next, the study performed by Mesa et al. [103] demonstrated that siderophores increased As accumulation in the roots and leaves of *Betula celtiberica*, where, in addition to As concentration, soil pH reactions via interaction with endophytes played an important role in As transfer from soil to plant. In the study by [104], the strain *nbri05* (isolated from As-contaminated soil) reduced the toxicity of As in chickpea plants by reducing As accumulation in shoots and promoting phytostabilization by accumulating higher levels of As in the root. In the study by [105], five different arsenate-reducing bacterial strains (*Rhodococcus* sp. TS1, *Delftia* sp. TS33, *Delftia* sp. TS41, *Streptomyces lividans* sp. PSQ22 and *Comamonas* sp. TS37) improved *Pteris vittate* biomass by 53%, As uptake by 44% and reduced As leaching in soil by 29% to 71%. In the study by [106], As-tolerant endophytic microbes isolated from the As hyperaccumulator *Lantana camara* improved the growth of *Solanum nigrum* with increased accumulation of As. In addition, the importance of endophytic strains was confirmed in upregulation of MRP transporter in the root, suggesting that the consortium of arsenic-tolerant endophytic strains altered the accumulation or detoxification of As process in *Solanum nigrum* by improving As phytoremediation.

## 9. Conclusions and Future Perspectives

Arsenic pollution in ecosystems has increasingly affected the health of many organisms, including humans; therefore, As management is of vital importance for environmental safety and security. Various conventional methods (chemical, physical and/or agro-hydrotechnical) have long been used to remediate As-contaminated media, but often with the limitations of high cost, low efficiency and/or toxicity. As a very promising alternative, three specific plant species groups, (i) As-resistant, (ii) As-accumulating and (iii) As-hyperaccumulating, exhibit a certain level of tolerance to As exposure, making them a sustainable option for As phytoremediation. In addition, various microbes, individually and/or in combination with certain plants, are also used for bioremediation of As contamination. Both approaches (bioremediation/phytoremediation) are cost-effective and environmentally friendly and can be applied as on-site methods, reducing the need for transport and disposal of contaminated material. Therefore, priority should be given to studies that focus on elucidating the genetic and enzymatic reactions and mechanisms involved in the uptake and translocation of As species in the plant tissues. Furthermore, bioremediation or phytoremediation can be used in combination with other methods such as physical (e.g., separation, deposition, mixing) and/or chemical (immobilization, solidification, adsorption) methods. However, it is also important to note that bio-/phytoremediation can be influenced by numerous separate and/or combined environmental variables (pH, temperature, presence of inorganic ligands, salt concentration). Therefore, it is important to consider site-specific conditions and adapt them to particular plant species, microbial strains, etc., in order to achieve optimal remediation results. For example, some of the recent studies show that the biochemical reactions of metalloids (Cd) as well as their metabolites and metabolic pathways under contaminated conditions can be significantly altered by the simultaneous application of (i) Cl salinity [107,108] or (ii) microplastic pollution [109]. However, multiple interactions between different environmental variables lead to multicollinearity, making specific and synergistic effects extremely complex to study, notably in non-controlled field conditions.

Continued advances in biotechnology, genetic engineering and material science have led to the development of promising genetically-improved organisms (microbial strains, plants) and novel organo-mineral materials (nano-biochar, nano-aluminosilicates) that can absorb and deposit As more efficiently, further contributing to the bioremediation of As. For instance, nanobiotechnology has strong outputs in developing and improving nano-sized particles (NPs) with higher active surface area, specific in shape and composition, which, in soil conditions, markedly impact As biogeochemistry (e.g., NPs with organic acids co-precipitate), reducing As mobility and phytoaccumulation [110]. In addition, recently, the authors of [111] developed novel organo-mineral composites (nano-zero valent Zn and biochar-hydroxyapatite-alginate) for an effective removal of As(III) and As(V) from the water matrix. Furthermore, nano-Fe_3_O_4_ and nano-zero valent Fe were confirmed to reduce As phytoaccumulation through an adsorption mechanism, while the ameliorative role of cerium oxide NPs in controlling As toxicity was confirmed, but the mechanism is not yet fully explained [110]. Therefore, further studies are needed to uncover the underlying mechanisms and optimize the use of such advanced materials and their application in different soil and water matrices for them to be efficient in As remediation. Finally, a possible realistic obstacle to achieving such goal(s) is the challenge of scaling up, i.e., remediation techniques that have proven successful at the lab scale may not be practical or cost-effective when transferred to a larger scale in the field.

## Figures and Tables

**Figure 1 molecules-28-01474-f001:**
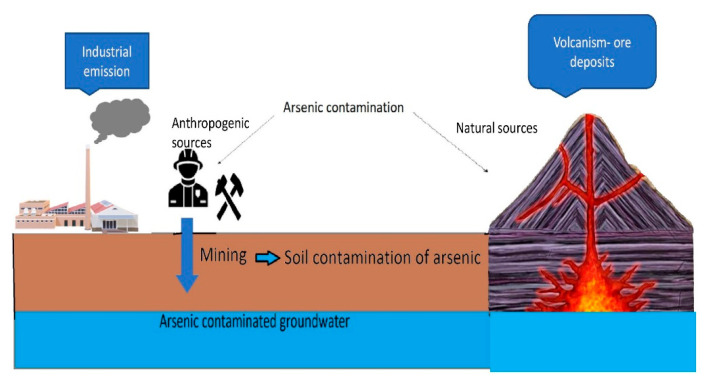
Sources of arsenic contamination in the environment.

**Figure 2 molecules-28-01474-f002:**
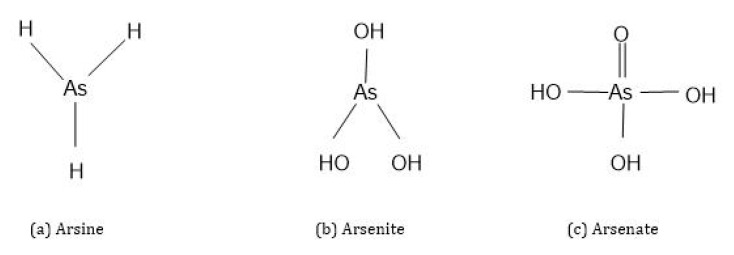
Chemical structures of inorganic arsenic species.

**Figure 3 molecules-28-01474-f003:**
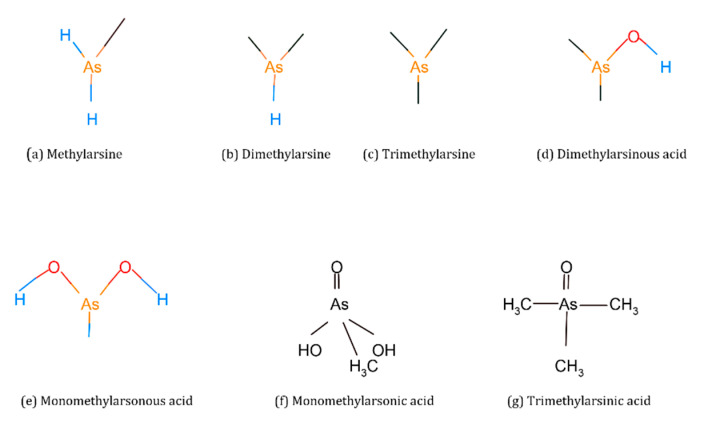
Chemical structures of organic arsenic species.

**Figure 4 molecules-28-01474-f004:**
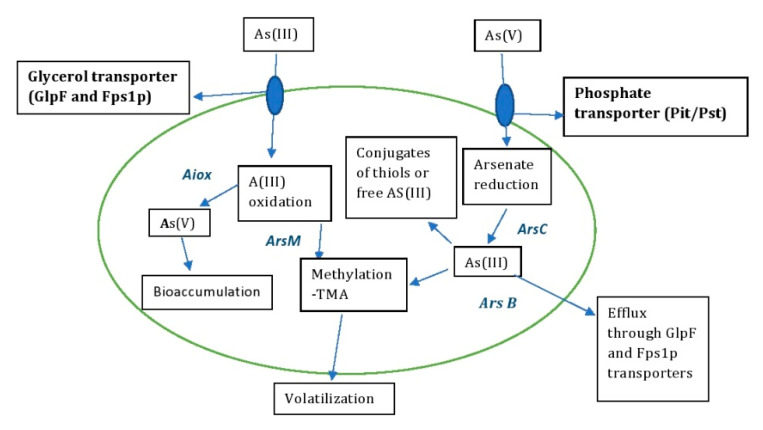
Arsenic detoxification mechanism by microbes—bioremediation.

**Figure 5 molecules-28-01474-f005:**
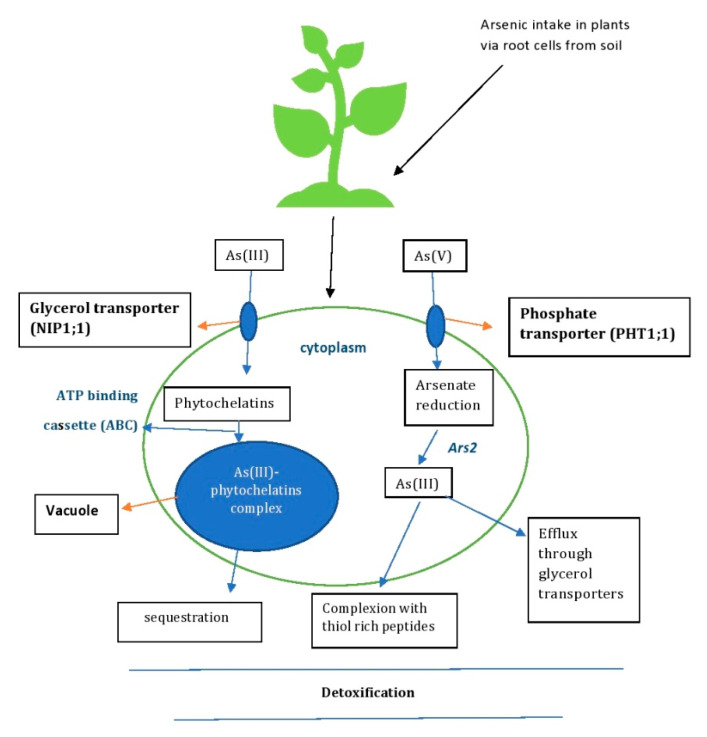
Mechanisms of As detoxification by plants—phytoremediation.

**Table 1 molecules-28-01474-t001:** Arsenic-tolerant microbial strains.

No.	Strain Isolate	Tolerance	Reference
1	*Brevundimonas aurantiaca* PFAB1	Up to 90 mM arsenite Up to 310 mM arsenate	[45]
2	*Bacillus* sp. *KM02*, *Aneurinibacillus aneurinilyticus*	Up to 4500 ppm arsenate Up to 550 ppm arsenite	[46]
3	*Bacillus* sp. *KL6*	Up to 90 ppm arsenic	[47]
4	*Alicyclobacillus mali* FL 18	Up to 41mM arsenite	[48]
5	*Aeromonas*	Up to 100 mM arsenate Up to 20 mM arsenite	[49]
6	*Microbacterium*	Up to 69.2 mM arsenite	[50]
7	*Pseudomonas* sp. QNC1	Up to 350 mM arsenate	[51]
8	*Lysinibacillus* sp., *Bacillus safensis*	Up to 88.53 mM arsenite Up to 721.13 mM arsenate	[52]
9	*Klebsiella pneumonia* sp.	Up to 28 mM of arsenic	[53]
10	*Brevibacterium linens* strain AE038-8	Up to 1 M of arsenate Up to 75 mM of arsenite	[54]
11	*Acromobacter xylosoxidans*	Up to 15 mM As(III)	[55]

**Table 2 molecules-28-01474-t002:** Arsenic-hyperaccumulating plant species.

Species	As Concentration in Tested Medium	As Accumulation in Test Plant	Reference
*Pteris vittata*	8885 ± 1640 mg/kg soil	7215 to 11,110 mg/kg	[75]
*Landolita punctata*	0.5–3.0 mg/L nutrient solution	>1000 mg/kg	[76,77]
*Eichhornia crassipes*	0.5 mg/L nutrient solution	498.4 mg/kg	[78]
*Azolla caroliniana*	0.25–1.5 mg/L nutrient solution	386.1 µg/g	[79]
*Hydrilla verticillata*	375 µg/L artificial fresh water	197.2 ± 17.4 µg/g	[80]
*Spirodela polyrhiza* L.	75–300 µg/L nutrient solution	26.4 ± 0.22 µg/g	[81]

## Data Availability

All data are available upon request.

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
