# Peer review of "Biotechnology Advances in Bioremediation of Arsenic: A Review"

_molecules, 2023, doi:10.3390/molecules28031474_

Round 1

Reviewer 1 Report

This is a very interesting and generally well-written review. The authors have covered the information well. A few suggestions have been made for some of the writing.

Abstract

Line 11; The authors start out the sentence with the word Arsenic but convert to using 'As' in the rest of the Abstract. They should make the change to 'Arsenic (AS)' to start the sentence. 'AS' is recommended instead of 'As' because the latter can be confused with the word 'as'.

Line 14; should change to 'methods and approaches are needed to prevent and protect humans and other organisms from detrimental AS exposure'

Line 15; change 'by using of microbes' to 'by using microbes'

Lines 16-17; change to 'considered as cost-effective'

Lines19-22; Change sentence to read 'In this review the most recent insights from AS bio/phyto-remediation are presented, and the most relevant physiological and molecular mechanisms involved in AS biological routes, that can be useful starting points in the creation of more AS tolerant microbes and plants, as well as their symbiotic associations are discussed.'

Main text

Line 40; Change to 'In the soluble form,'

Line 43-44; Change to 'in the food chain by plant uptake and food crop consumption'

Lines 45-46; Change to 'The main sources of AS exposure for humans are use of contaminated water for drinking/cooking, growing crops in'

Lines 134-135; Change to 'Arsenite oxidase, synthesized by the bacterial species'

Lines 182-183; Change to 'detoxification process of R. palustris'

Lines 199; Change to 'Plants take up heavy metals dominantly via the root system'

Lines 252-253; Change to 'For instance, an arsM gene'

Lines 267-268; Change to 'In the case of [88], PCR analyses of bacterial samples isolated from different AS-contaminated zones'

Lines 287; Do the authors mean 'ArsR1 and ArsR2 in Pseudomonas putida'?

Line 288; Do the authors mean 'The crosstalk between the ArsRs was investigated'

Line 298; Change to 'and are discussed in the next paragraphs.'

Line 299; Change to 'For instance, in the study performed'

Line 353; Change to 'metal-contaminated zones interact with'

Line 364; Change 'microbiotal' to 'microbiota'

Lines 376-377; Change to 'suggesting that the consortium of arsenic tolerant endophytic strains'

Author Response

Please find enclosed our revised version R1 of the Manuscript.

The manuscript R1 version has been checked and improved according to your comments and suggestions with active track change option. Our responses (red text) to the comments (black text) are presented in the next. Please see in the attachment.

Reviewer 2 Report

In this manuscript, the authors review the latest research advances in As bioremediation and discuss the most relevant physiological and molecular mechanisms in the As biopathway. The whole article is relatively comprehensive, and there are some details that need attention. Major revisions are recommended.

1.    The logic of the introduction is not tight enough. It is suggested to reorganize it. In addition, the last paragraph of the introduction should make a small summary of the entire article.

2.    More illustrations, such as Figure, are needed to make the discussions more visualized.

3.    The table recommends using a three-line table.

4.    The review should end with a vision of the future of the topic (potential problems, potential challenges, etc.).

5.    The format of the references is confused, with mixed Chinese and English.

6.    The conclusion part should present key conclusions of bioremediation of arsenic.

Author Response

(The authors gave the same response as above.)

Reviewer 3 Report

Manuscript Number: 2154410

The manuscript entitled “Biotechnology advances in bioremediation of arsenic: A review”. This review is very promising and well written. Bioremediation of heavy metal is one of the major concern all over the world.  So, this review report is very interesting and suitable for this journal.

Comments

1.     In the introduction part “Contamination of soils and waters with arsenic (As) as a highly toxic element is an increasing environmental issue….”. You can rewrite this sentence. (lane27)

2.     In introduction you can describe arsenic pollution  from step by step like, first arsenic in its natural form in the environment, then possible toxicities, current report toxicities in global, Asia..etc.

3.     In the lane 86, mentioned about arsenic related toxicities on different vital organs like lung, kidney, heart and liver. So you can incorporate a diagrammatic representation of vital organ toxicity also.

4.     In the session of Molecular mechanisms involved in As microbial remediation. You can start the first paragraph with significance of metal microbe interactions in two sentences. Rest of the paragraph is well narrated.

5.     In lane 188, Biosorption and bioaccumulation of As. You can start with lane 195 (last sentence in that para).

6.     In lane 261, start with “ The genetically modified strain Bacillus subtilis… simply this sentence

7.     In lane 321, “In the phytoremediation strategy of [98], this sentence also you want to simplify.

8.     Over all this review is well written in a simple manner with lot of information.

The manuscript is recommended for acceptance after this minor modification.

Author Response

(The authors gave the same response as above.)
